# Evaluation of the New Individual Fatty Acid Dataset for UK Biobank: Analysis of Intakes and Sources in 207,997 Participants

**DOI:** 10.3390/nu14173603

**Published:** 2022-08-31

**Authors:** Rebecca K. Kelly, Zoe Pollard, Heather Young, Carmen Piernas, Marleen Lentjes, Angela Mulligan, Inge Huybrechts, Jennifer L. Carter, Timothy J. Key, Aurora Perez-Cornago

**Affiliations:** 1Cancer Epidemiology Unit, Nuffield Department of Population Health, University of Oxford, Oxford OX3 7LF, UK; 2Nuffield Department of Primary Care Health Sciences, University of Oxford, Oxford OX3 7LF, UK; 3Department of Public Health & Primary Care, Institute of Public Health, University of Cambridge, Cambridge CB1 8RN, UK; 4School of Medical Sciences, Clinical Epidemiology and Biostatistics/Nutrient Gut Brain Interaction, Örebro University, 70182 Örebro, Sweden; 5Nutrition Measurement Platform, MRC Epidemiology Unit, University of Cambridge, Cambridge CB2 0QQ, UK; 6International Agency for Research on Cancer, World Health Organization, 69372 Lyon, France; 7Clinical Trial Service Unit and Epidemiological Studies Unit, University of Oxford, Oxford OX3 7LF, UK

**Keywords:** food composition database, nutrient database, online 24 h dietary assessment, Oxford WebQ, fatty acids

## Abstract

The Oxford WebQ is an online 24 h dietary assessment tool used by several large prospective studies. This study describes the creation of the new individual fatty acid (FA) dataset for the Oxford WebQ and reports intakes and sources of dietary individual FAs in the UK Biobank. Participants who completed ≥1 (maximum of five) 24 h dietary assessments were included (*n* = 207,997). Nutrient intakes were obtained from the average of all completed 24 h dietary assessments. Nutrient data from the UK McCance and Widdowson’s The Composition of Foods and the US Department of Agriculture food composition tables were used to calculate intakes of 21 individual FAs. The individual FA dataset included 10 saturated fatty acids (SFAs), 4 monounsaturated fatty acids (MUFAs), and 7 polyunsaturated fatty acids (PUFAs; including alpha-linolenic (18:3), eicosapentaenoic (20:5), and docosahexaenoic (22:6) acids). Palmitic (16:0; mean ± standard deviation (SD): 13.5 ± 5.7 g/d) and stearic (18:0; 5.2 ± 2.5) acids were the main contributors to SFAs, and the main sources of these were cereals and cereal products (mostly desserts/cakes/pastries), milk and milk products (mostly cheese and milk), and meat and meat products. Oleic acid (18:1; 24.2 ± 9.8) was the main MUFA, derived mainly from cereals and cereal products, and meat and meat products. Linoleic acid (18:2; 9.7 ± 4.3) was the main PUFA, derived mostly from cereals and cereal products, and vegetables (including potatoes) and vegetable dishes. The individual FA dataset for the Oxford WebQ will allow future investigations on individual FAs and disease risk.

## 1. Introduction

Fatty acids (FAs) have structural and physiological functions in the human body [1]. The properties of FAs vary depending on their carbon-chain length and the presence, number, position, and configuration of any double bonds [1]. While total saturated fatty acids (SFAs), monounsaturated fatty acids (MUFAs), and polyunsaturated fatty acids (PUFAs) have been differently associated with various health outcomes [2,3,4,5], the evidence for individual FAs remains limited.

Evidence from recent observational studies suggests that the relationship between SFAs and cardiovascular disease (CVD) may depend on their carbon-chain length. Higher intakes of palmitic acid (16:0) and stearic acid (18:0) have been associated with higher [6,7,8,9] or neutral [10,11] risk of CVD, whereas short- to medium-chain SFAs (sum of butyric (4:0) through capric (10:0) acids) and odd-chain SFAs (e.g., pentadecanoic (15:0) and heptadecanoic (17:0) acids) have been associated with neutral [6,7,8] or lower [9,10,11] risk and associations of lauric (12:0) and myristic (14:0) acids with risk are inconsistent [6,7,8,9,10,11]. Randomised controlled trials (RCTs) have found that lauric (12:0), myristic (14:0) and palmitic (16:0) acids increase low-density lipoprotein cholesterol (LDL-C), while the effect of stearic acid (18:0) on LDL-C is largely neutral, and evidence is lacking for the effect of short- to medium-chain and odd-chain SFAs [12,13]. However, these associations remain unclear in observational studies, and evidence is limited particularly for individual MUFAs and PUFAs and for other health outcomes.

The Oxford WebQ is a self-administered 24 h online dietary assessment tool used to measure dietary intakes in several large observational studies [14], including the UK Biobank [15] and the Million Women Study [16]. Nutrients are estimated automatically from the participants’ responses using built-in algorithms and data from food composition tables (FCTs) [17]. Data on individual FAs are not available in most FCTs, including the UK Nutrient Databank (UKNDB) used to calculate the main dataset for the Oxford WebQ [18]. Therefore, the McCance and Widdowson’s the Composition of Foods Integrated Dataset (CoFID) [19] and the US Department of Agriculture (USDA) [20] FCTs, which both contain data on individual FAs, were used to calculate intakes in the individual FA dataset.

The aim of this study was to describe the calculation of the new individual FA dataset for the Oxford WebQ and to report dietary intakes and food sources of individual FAs in 207,997 UK Biobank participants. Our secondary aim was to compare the estimation of major nutrient intakes (total energy, fats and fat subtypes, carbohydrates, and protein) between the new individual FA dataset and the main dataset.

## 2. Materials and Methods

### 2.1. Study Design

The UK Biobank includes 210,969 middle aged men and women (42% of whole cohort) who completed at least one 24 h dietary assessment (Oxford WebQ) between 2009 and 2012 [21]. Participants were recruited across 22 assessment centres in England, Wales, and Scotland, and detailed information on lifestyle, environmental and health factors, and biological samples and anthropometric measurements were collected at baseline. All participants provided informed consent and approval was obtained from the National Information Governance Board for Health and Social Care and the National Health Service North West Multicentre Research Ethics Committee (reference number 21/NW/0157, approval letter dated 29 June 2021). The UK Biobank study protocols and information on data access for researchers are located online (https://www.ukbiobank.ac.uk/media/gnkeyh2q/study-rationale.pdf (accessed on 10 August 2022)) [21].

### 2.2. Study Participants

Participants who were recruited in the UK Biobank study between April 2009 and September 2010 completed the Oxford WebQ in the assessment centres at baseline [21]. Further, all participants who provided a valid email address were invited via email to complete the Oxford WebQ on up to four occasions during the follow-up period between February 2011 and June 2012. Participants were excluded if they did not have at least one valid 24 h dietary assessment; dietary assessments with implausible energy intakes (outside the range of 3347 to 17,573 kJ/d, or 800 to 4200 kcal for men, and 2092 to 14,644 kJ/d, or 500 to 3500 kcal for women) [22] calculated with either dataset (individual FA dataset or main dataset) were removed (*n* = 2972 participants; see Appendix A). In these analyses, we were interested in describing nutrient intakes in the overall study population (not individual participant intakes), and therefore, we have included participants with only one 24 h dietary assessment. This study included 207,997 participants (out of 210,969, 99%) who completed between one and five 24 h dietary assessments as follows: 1 (*n* = 83,906, 40.3%), 2 (*n* = 48,316, 23.2%), 3 (*n* = 41,799, 20.1%), 4 (*n* = 28,725, 13.8%) and 5 (*n* = 5251, 2.5%).

### 2.3. Dietary Assessment Using the Oxford WebQ Questionnaire

The Oxford WebQ is an online 24 h dietary assessment tool that collects information on up to 206 foods and 32 beverages consumed during the previous day (https://biobank.ctsu.ox.ac.uk/crystal/ukb/docs/DietWebQ.pdf (accessed on 10 August 2022), UK Biobank Showcase category: 100090) [14,17]. Nutrient intakes for each food or beverage are obtained by multiplying the amount consumed (using the portion size and daily frequency of intake) by the nutrient composition obtained from FCTs. The calculation of portion sizes for the Oxford WebQ is described elsewhere [17]. Nutrient intakes estimated from the Oxford WebQ using earlier versions of the McCance and Widdowson FCT (2002) were considered to perform moderately well as estimates of true intakes when validated against total energy expenditure (estimated from accelerometery), a recovery biomarker for protein intake, and a predictive biomarker for sugar intake [23], and were similar to those obtained from interviewer-administered 24 h dietary recalls [14].

### 2.4. Individual Fatty Acid Calculation for the Oxford WebQ Questionnaire

McCance and Widdowson’s The CoFID 2021 [19] (subsequently referred to as ‘MCW’) was the primary FCT used to calculate individual FA intakes because this is the only UK-specific FCT with individual FA data. The USDA National Nutrient Database for Standard Reference release 26 2013 [20] (subsequently referred to as ‘USDA’) FCT was used if individual FA data were not available in MCW. We selected the USDA as the secondary FCT because it included comprehensive individual FA data for a large number of foods (~8000 food codes available) and there was a version compiled in 2011–2012, which was close in time to the completion of the Oxford WebQ in UK Biobank (2009–2012).

#### 2.4.1. Food Code Matching

Food and beverage items in the Oxford WebQ have already been matched to UKNDB food codes for the main dataset [17], and these UKNDB food codes were matched to MCW food codes for the individual FA dataset (see Appendix A). The matched MCW food codes that may be important sources of individual FAs (i.e., >0.5 g of total fat per 100 g of food) were reviewed. We identified 184 matched MCW food codes that were missing individual FA values (including values listed as ‘N’ to signify where a nutrient is present in significant quantities but there is no reliable information on the amount) and 8 matched MCW food codes with implausible values for individual FAs (i.e., the sum of individual SFAs, MUFAs or PUFAs was ≥1.2 times higher than the total SFAs, MUFAs or PUFAs reported in MCW, respectively; see Appendix A). At this stage, we identified two MCW food codes with implausible individual FA values due to probable data entry errors in MCW, and these values were replaced by tracing back to the original data source. In accordance with the Food and Agriculture Organization of the United Nations (FAO) FCT guidelines [24], the remaining 190 food codes were replaced following these chronological steps and prioritising the use of UK-based MCW food codes where possible:

1Food codes were replaced with similar food codes from MCW (e.g., English cheddar cheese was used instead of Caerphilly cheese) or a different form of the same food code from MCW (e.g., raw herring with flesh was used instead of grilled herring no bones). Food codes were considered similar if they had similar descriptions and similar macronutrient (including SFA, PUFA and MUFA) content per 100 g of food to the MCW food code being replaced. In total, 103 food codes were replaced with similar MCW food codes.2If step 1 above could not be applied because there was no closely similar food code with FA data, we replaced the food code with a composite of several MCW food codes in a recipe based on ingredients from industry and online recipes. For example, we replaced Waldorf salad (food code: 15-878) using 45% raw apples (14-326), 30% raw celery (13-636), 15% mayonnaise (17-654), and 10% walnuts (14-879). In total, 42 food codes were replaced with recipes using MCW food codes.3If neither step 1 nor step 2 could be applied, we used similar food codes from USDA, and if we were unable to replace the food code with a similar USDA food code, we replaced the food code with a composite of MCW and USDA food codes in a recipe. In total, 40 food codes were replaced with similar USDA food codes, and 5 food codes were replaced with recipes using both MCW and USDA food codes.

The final individual FA dataset comprised 437 unique food codes from the FCTs, including 406 food codes from MCW (93%) and 31 food codes from USDA (7%). In some instances, a single food code from MCW or USDA was used to replace several different MCW food codes; therefore, the total number of unique food codes in the individual FA dataset is lower than the number of MCW food codes that were replaced. See Appendix A for the food codes assigned to WebQ items in the individual FA dataset, and Appendix A for a detailed list of the USDA food codes incorporated into the individual FA dataset.

#### 2.4.2. Quality Assessment

Five researchers were involved in the quality assessment of the individual FA dataset. Two researchers (researcher 1/food scientist, researcher 2/nutritionist) matched the UKNDB food codes used in the main dataset to MCW foods codes. Inconsistencies were discussed and a third researcher (researcher 3/senior nutritional epidemiologist) reviewed all the food code matching and suggested changes to the selected food codes and the fractions assigned to each food code within each WebQ item, and further modifications were made after discussion with the other researchers (researcher 1, researcher 2). Another researcher (researcher 4/PhD candidate in nutritional epidemiology) identified matched MCW food codes with missing individual FAs or implausible values for individual FAs and two researchers (researcher 1, researcher 4) identified replacements for these food codes and inconsistencies were discussed and resolved by consensus (see Section 2.4.1 above). Three researchers (researcher 1, researcher 3, researcher 4) reviewed the food code matching and fractions assigned to each food code within each WebQ item and any modifications were discussed and agreed upon by all. Each WebQ item in the individual FA dataset was checked twice and compared with the amounts in the main dataset to ensure that the fractions for each food code within each WebQ item totalled 100% (researcher 1, researcher 3, researcher 4). Researcher 5 (statistical programmer) compiled the individual FA dataset, performing a final check and verifying that there were no missing FA values for any food code, and that the fractions assigned to food codes within each WebQ item totalled 100%.

The individual FA dataset incorporated 21 individual FAs (see Appendix A), including 10 individual SFAs (butyric acid (4:0), caproic acid (6:0), caprylic acid (8:0), capric acid (10:0), lauric acid (12:0), myristic acid (14:0), pentadecanoic acid (15:0), palmitic acid (16:0), heptadecanoic acid (17:0), stearic acid (18:0)), 4 individual MUFAs (palmitoleic acid (16:1), oleic acid (18:1), eicosenoic acid (20:1), docosenoic acid (22:1)) and 7 individual PUFAs (linoleic acid (LA; 18:2), alpha-linolenic acid (ALA; 18:3), stearidonic acid (18:4), arachidonic acid (20:4), eicosapentaenoic acid (EPA; 20:5), docosapentaenoic acid (DPA; 22:5), and docosahexaenoic acid (DHA; 22:6)) (see Appendix A). Because values for some individual FA isomers were sometimes missing from the MCW or USDA FCTs, the values for the total undifferentiated individual FA were used. Total dietary linolenic acid is mostly ALA; therefore, values for total linolenic acid were used as a surrogate for ALA in the individual FA dataset [25,26]. Moreover, we also incorporated total energy and major nutrients (fat, total SFAs, total MUFAs, total PUFAs, trans fatty acids (TFAs), carbohydrates, and protein) from MCW/USDA into the individual FA dataset for comparison with the main dataset. TFAs are also included in total MUFAs and total PUFAs [19]. Total *n*-3 PUFAs and total *n*-6 PUFAs were not available for some MCW food codes and were not available for any USDA food codes; therefore, we calculated total *n*-3 PUFAs (sum of ALA (18:3), stearidonic acid (18:4), EPA (20:5), DPA (22:5) and DHA (22:6)) and *n*-6 PUFAs (sum of LA (18:2) and arachidonic acid (20:4)) based on the typical isomer of individual PUFAs for comparison with the main dataset.

### 2.5. Statistical Analyses

Selected baseline characteristics of participants included in this study were reported, including age at recruitment, sex, ethnicity, education, alcohol intake, smoking status, physical activity, and body mass index (BMI). Further details regarding the assessment of participant characteristics at baseline are published elsewhere [27]. We calculated participants’ nutrient intakes from the average of their completed 24 h dietary assessments, and we then calculated the mean, standard deviation (SD), median, interquartile range (IQR), 5th and 95th percentiles of nutrient intakes. Food and beverage items in the individual FA dataset were classified into 93 food groups and aggregated into 14 primary food groups and 53 secondary food groups based on the UK National Diet and Nutrition Survey (NDNS) classification, which were then used to calculate dietary sources of individual FA intakes (see Appendix A) [28].

The absolute and percentage differences in nutrient intake estimations between the individual FA dataset and the main dataset were calculated, and the means were compared using paired *t* tests for normally distributed nutrients and Wilcoxon’s rank sum test for non-normally distributed nutrients (i.e., total PUFAs, *n*-3 PUFAs, *n*-6 PUFAs, TFAs). Spearman correlation coefficients were calculated to compare intakes of total energy and major nutrients estimated in the individual FA dataset and the main dataset. Participants were categorised by fifths of nutrient intake, and we calculated both the proportions of participants in the same or adjacent fifth and the weighted Kappa statistics to compare the rankings of nutrient intakes estimated in the individual FA dataset and the main dataset. We interpreted weighted Kappas as follows: none (≤0), none to slight (0.01–0.20), fair (0.21–0.40), moderate (0.41–0.60), substantial (0.61–0.80) or almost perfect (≥0.81) agreement [29].

STATA version 17.0 (Stata Corporation, College Station, TX, USA) was used for all data analyses.

## 3. Results

Participants had a mean age at recruitment of 56.1 (SD 7.9) years, 55% were women, 96% reported being of white British ancestry, 77% reported having a higher education degree, 8% were current smokers, and 15% reported high levels of physical activity (see Appendix A). Participants’ mean BMI was 26.9 (SD 4.6) kg/m^2^, and mean alcohol intake excluding non-drinkers (~6.2%) was 16.3 (SD 17.2) g/d.

### 3.1. Intakes of Individual Fatty Acids

Intakes of individual FAs in UK Biobank calculated from the individual FA dataset are displayed in Table 1. The main contributors to total SFA intakes (in descending order) were palmitic acid (16:0; mean ± SD g/d: 13.5 ± 5.7), stearic acid (18:0; 5.2 ± 2.5), myristic acid (14:0; 2.4 ± 1.3), and lauric acid (12:0; 1.2 ± 0.7). The individual FA contributing most to total MUFA intakes was oleic acid (18:1; 24.2 ± 9.8), while the main contributor to total PUFA intakes was LA (18:2; 9.7 ± 4.3). Intakes of most individual FAs were higher in men compared with women.

The top five food groups contributing to palmitic acid (16:0), stearic acid (18:0), oleic acid (18:1) and LA (18:2) intakes are shown in Figure 1. From 14 primary food groups, we observed that the top contributors to palmitic acid (16:0) and stearic acid (18:0) were cereals and cereal products (27.7% and 18.3%, respectively; mostly desserts/cakes/pastries), milk and milk products (22.1% and 23.9%; mostly cheese and milk), and meat and meat products (14.6% and 21.3%). The top contributors to oleic acid (18:1) were cereals and cereal products (22.0%) and meat and meat products (15.7%), while the top contributors to LA (18:2) were cereals and cereal products (25.3%), and vegetables (including potatoes) and vegetable dishes (15.2%). Food groups contributing to all individual SFAs, MUFAs and PUFAs are shown in Appendix A, respectively.

### 3.2. Comparison of Major Nutrient Intakes in the Individual Fatty Acid Dataset and Main Dataset

Table 2 displays the mean, median, 5th and 95th percentiles for total energy and major nutrients for the main dataset as compared with the individual FA dataset, as well as the mean differences in intakes between the two versions. Although we found significant differences between the means of intakes in both versions, these were not large. Compared with the main dataset, intakes in the individual FA dataset were slightly lower for total energy and all major nutrients except for total MUFAs, which were slightly higher, although percentage difference in means between both versions were all <10%, except for TFAs (−12.0%).

Spearman correlations and strength of agreement on ranking total energy and nutrient intakes between the main and individual FA dataset are displayed in Table 3 and Appendix A. We observed high correlations (*r* > 0.86) between the same nutrients estimated in the two datasets. Overall, agreement between both datasets was high, with most nutrients classified in the same or adjacent fifth ranging from 93.8% for *n*-3 PUFAs (*κ* 0.69) to 100.0% for total energy (*κ* 0.95). The ranges of intakes within each fifth are found in Appendix A and the food sources of fat subtotals between the main dataset and individual FA dataset are found in Appendix A.

## 4. Discussion

This study described individual FA intakes and their main food sources in the new individual FA dataset for the Oxford WebQ 24 h dietary assessment and compared intakes of major nutrients with those from the main dataset (UKNDB) among participants in the UK Biobank. We incorporated 21 individual FAs into this new dataset, including 10 individual SFAs, 4 individual MUFAs, and 7 individual PUFAs. We also included total energy and 7 macronutrients in the individual FA dataset for comparison with the main dataset; there were only small absolute differences in the means of these nutrient intakes between the two datasets (with the exception of TFAs), and the ranking of individuals was minimally affected.

To the best of our knowledge, this is the only large cohort study to report intakes of individual FAs in the UK, with the exception of the European Prospective Investigation into Cancer (EPIC)-Norfolk [11], which used individual FA data from earlier McCance and Widdowson FCTs (1998) to calculate intakes for 22,050 participants between 1993 and 1997 [30]. Palmitic acid (16:0) and stearic acid (18:0) were the main contributors to SFAs in our study (52.3% and 20.2%, respectively). Intakes of palmitic acid (16:0) and stearic acid (18:0) estimated from food frequency questionnaires in EPIC-Norfolk (~54.6 and ~22.2%) [11] and among US adults aged ≥20 years in the National Health and Nutrition Examination Survey (NHANES) 2011–2012 (~54.0 and ~24.8%) [31] accounted for a slightly higher proportion of SFAs. Moreover, short-chain SFAs (sum of butyric (4:0) through capric (10:0) acid), accounted for a higher proportion of total SFA intakes in the EPIC-Denmark cohort (~8.7%) [11], and a lower proportion of total SFAs intakes in the EPIC-Norfolk cohort (~5.5%) [11] and NHANES 2011–2012 (~6.0%) [31], in comparison to the current study (7.9%). The proportion of SFAs from odd-chain SFAs (sum of pentadecanoic (15:0) and heptadecanoic (17:0) acids), was similar between the UK Biobank (2.1%) and EPIC-Norfolk (~2.4%) [11]. Longer-chain SFAs (e.g., palmitic and stearic acids) are mainly found in meat, dairy and vegetable fats/oils, while short-chain and odd-chain SFAs are mainly found in dairy; therefore, differences in intakes of these foods between populations in the UK, US and Denmark may partly explain differences in individual SFA intakes (e.g., higher meat and lower dairy intakes in the US versus UK) [32]. Meat and meat products accounted for a larger proportion of SFA intakes among US adults aged ≥20 years in NHANES 2005–2006 (~24.0%) [33] and UK adults aged 19–65 years in the 2008–2012 NDNS (~24.5%) [34] as compared with EPIC-Norfolk (~12.7%) [11] and the current study (13.5%). Milk and milk products accounted for a higher proportion of SFA intakes in this study (27.7%) and EPIC-Norfolk (~29.4%) [11] and EPIC-Denmark (~29.9%) [11] as compared with US adults aged ≥20 years in NHANES 2005–2006 (~21.4%) [33] and UK adults aged 19–65 years in the 2008–2012 NDNS (~22.0%) [34]. Differences in the dietary assessment methods (e.g., 4-day food diary in NDNS) and the FCTs used to calculate SFA intakes (e.g., UKNDB in NDNS) may also account for some of the differences in individual FA intakes observed between populations. Moreover, dietary patterns and nutrient intakes in UK Biobank may not be representative of the general UK population [15,35].

In our study, the major contributor to MUFAs was oleic acid (18:1; 24.2 g/d), which was slightly lower than oleic acid (18:1) intake in US adults ≥20 years adults in NHANES 2011–2012 (27.5 g/d) [31]; as far as we are aware there are no other data for individual MUFAs in the UK for comparison. However, total MUFA intakes in our study (26.8 g/d) were similar to those reported for the EPIC-Norfolk study (men 55–64 years, 29.6 g/d; women 55–64 years, 20.4 g/d) [36] and adults aged 19–64 years in NDNS 2011–2012 (25.1 g/d) [34], while intakes in southern European EPIC sites (Italy, Greece and Spain) were higher (range for men and women 55–64 years, 29.5 to 54.3 g/d) [36]. Food sources of MUFAs differed by cohort site in EPIC, which may explain differences in intakes; olive oil was the primary food source in southern European EPIC sites (~40% of total MUFAs) [36], while cereals and cereal products and meat and meat products were the main food sources in UK Biobank and other central and northern European EPIC sites [36]. However, it was not possible to determine the amount of olive oil consumed in this study due to the way these questions were asked in the questionnaire. Instead, a standard percentage of fat (where the fat/oil type used was selected by the participant) was added to vegetables usually cooked with some fat [17].

While LA (18:2) was the main contributor to PUFAs in this study, intakes of LA (18:2, 9.7 g/d) were 7.7 g/d lower than those estimated among US adults aged ≥20 years in NHANES 2011–2012 (17.4 g/d) [31]. However, intakes of *n*-6 PUFAs in our study (9.8 g/d), which were mostly derived from LA (18:2) (~98.7%), were similar to those reported for adults aged 19–64 years in the NDNS 2008–2012 (10.0 g/d) [34]. Cereals and cereal products (mainly cakes, desserts and pastries) and vegetables (including potatoes) and vegetable dishes (mostly from fats/oils added in cooking) were the primary sources of LA (18:2) in this study, NDNS 2008–2012 [34], and US adults aged ≥20 years in NHANES 2005–2006 [33]. Intakes of ALA (18:2) in this study (1.58 g/d) were also lower than those reported for US adults aged ≥20 years in NHANES 2011–2012 [31] (1.82 g/d). The typical fats and oils added to foods during cooking and processing differ between countries, which may partly explain the differences in FA intakes observed: soybean oil (54 g 18:2 per 100 g; 7.8 g 18:3 per 100 g) accounts for ~50% of oil consumed in the US [33,37], and palm oil (10.4 g; 0.2 g), rapeseed oil (20 g; 9.8 g) and sunflower oil (67 g; 0 g) together account for ~76% of oil consumed in the general UK population [38], while olive oil (10.6 g; 0.6 g) was the most common fats/oil used during cooking in the UK Biobank. Moreover, intakes of DHA (22:6; 0.18 g/d) in the present study were higher than those estimated for adults aged ≥20 years NHANES 2011–2012 [31] (0.06 g/d), while intakes of EPA were reported to be the same (20:5; 0.03 g/d). The type and amount of seafood, particularly oily fish [25], consumed may account for some of the between-country differences observed in EPA (20:5) and DHA (22:6) intakes: tuna (0.01 g 20:5 per 100 g; 0.08 g 22:6 per 100 g), cod (0.03 g; 0.10 g) and salmon (0.93 g; 1.78 g) were the most commonly consumed seafoods in the UK [19,39], while shrimp (0.09 g; 0.09 g) and to a lesser extent tuna and salmon were the most commonly consumed seafoods in the US [20,40].

Further, there was very good agreement between absolute intakes of total energy and major macronutrients estimated from the main dataset and individual FA dataset, although TFAs were lower (−0.14 g/d (−12.0%)), which may be because the food industry has voluntarily reduced or eliminated artificial TFAs in processed foods in recent decades and the MCW data are, on average, more recent than the UKNDB data [41]. Importantly, ~57% of the UKNDB food codes (for the main dataset) were matched to exact MCW food codes (for the individual FA dataset), which is similar to the proportion of EPIC foods matched to exact MCW food codes (~60%) for the EPIC Nutrient Databank for EPIC-UK sites [42]. Missing data in FCTs are an important source of error that may impact estimates of nutrient intakes [43,44]; individual FA values were published for only 706 (~24.5%) of the foods codes in MCW; therefore it was necessary to use another FCT (USDA) to replace some food codes. We expect some differences between UK and US FCTs due to environmental differences (e.g., soil, plant and livestock genetics, climate, livestock feed), differences in production and processing (e.g., storage conditions, fortification or enrichment, added oils and fats), and laboratory analyses (e.g., high-performance liquid chromatography, gas chromatography) [24,45,46]. Specifically, fat intakes between the main and individual FA datasets might differ for food items that include USDA food codes rich in added fats and oils from USDA (i.e., food codes for clarified butter ghee, light mayonnaise, regular cut white potato chips), due to differences in the typical oils and fats added in food cooking and processing, or milk and milk products from USDA (e.g., lactose reduced 2% fat cow’s milk, powdered milk not reconstituted), or due to geographical differences in livestock breed and feeding practices [47,48]. However, most of the USDA food codes we incorporated into the new dataset were fruits (*n* = 2), vegetables (with no fats/oils added in cooking; *n* = 23), or nuts and seeds (*n* = 2), for which we would not expect large between-country differences [47,48].

The individual FA dataset is separate to the main dataset and researchers examining major macronutrients (including fat subtotals) and micronutrients are advised to use the main dataset [17], although some researchers examining individual FA exposures alongside other nutrients may choose to rescale individual FAs to the fat subtotals in the main dataset. Because single 24 h dietary assessments are unlike to reflect usual intakes [23], researchers using this resource are advised to use a minimum of two 24 h dietary assessments to assess individual FA intakes.

This study has several strengths and limitations. To the best of our knowledge, this is the only open-access dataset on individual FAs available for a large prospective cohort study. The individual FA dataset was compiled using a rigorous process that followed international standards based on the FAO guidelines for food composition data [24]. While there may be differences in food composition between the UK and US, such as typical fats and oils added during cooking and processing, importation and exportation within the global food system is common and matching with non-UK data (i.e., USDA) is a scientifically valid solution for missing values that has been used in previous large prospective studies such as EPIC Europe [42,49]. ALA values were not always available in the FCTs; therefore, this dataset uses total linolenic acid values as a surrogate for ALA because most total linolenic acid is ALA [25], and researchers using this dataset in the UK Biobank may choose to do the same, as has been done in other large cohort studies [26,50,51,52,53]. USDA values for TFAs and odd-chain FAs (i.e., pentadecanoic (15:0) and heptadecanoic (17:0) acids) are not complete [20]; however, most food codes (~93%) were obtained from MCW. As with all dietary assessments based on self-reported questionnaires, estimates of nutrient intakes are prone to measurement error, particularly under- or over-reporting of dietary intakes [23]. The laboratory analyses used to extract and measure the individual FA composition of foods are also subject to some error [54].

## 5. Conclusions

In conclusion, we have described the calculation of 21 individual FAs and 8 major nutrients for a new individual FA dataset for the Oxford WebQ questionnaire and, for the 8 major nutrients, compared this with the main dataset used in UK Biobank; there were only small absolute mean differences between the two datasets and ranking was minimally affected. This resource will allow UK Biobank researchers to use these estimates of individual FA intakes for participants to examine the relationships between individual FA intakes and disease risk. The individual FA dataset will be returned to the UK Biobank and will be accessible alongside the main dataset.

## Figures and Tables

**Figure 1 nutrients-14-03603-f001:**
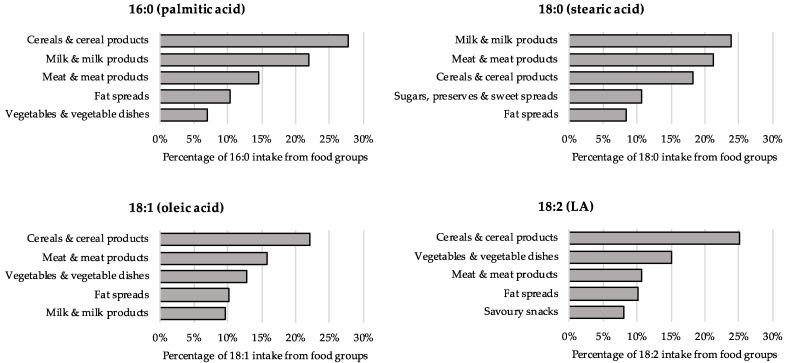
Top five food contributors to major individual FAs in the individual FA dataset (MCW + USDA) in 207,997 UK Biobank participants. FA, fatty acid; g/d, grams per day; LA, linoleic acid; MCW, McCance and Widdowson; MUFA, monounsaturated fatty acid; PUFA, polyunsaturated fatty acid; SFA, saturated fatty acid; USDA, US Department of Agriculture. Major individual FAs were those with mean intakes ≥5 g/d.

**Table 1 nutrients-14-03603-t001:** Intakes of individual FAs incorporated into the new individual FA dataset (MCW + USDA) in 207,997 UK Biobank participants.

Individual FAs (g/d), Mean (SD)	Total *n* = 207,997	Women *n* = 114,406	Men *n* = 93,591
SFAs			data
4:0 (butyric acid)	0.67 (0.45)	0.64 (0.41)	0.71 (0.48)
6:0 (caproic acid)	0.42 (0.28)	0.40 (0.26)	0.44 (0.30)
8:0 (caprylic acid)	0.34 (0.19)	0.32 (0.18)	0.36 (0.21)
10:0 (capric acid)	0.59 (0.36)	0.56 (0.34)	0.62 (0.39)
12:0 (lauric acid)	1.20 (0.69)	1.13 (0.64)	1.30 (0.74)
14:0 (myristic acid)	2.41 (1.27)	2.27 (1.16)	2.58 (1.38)
15:0 (pentadecanoic acid)	0.27 (0.16)	0.26 (0.14)	0.30 (0.17)
16:0 (palmitic acid)	13.45 (5.66)	12.44 (5.01)	14.70 (6.15)
17:0 (heptadecanoic acid)	0.28 (0.15)	0.26 (0.13)	0.31 (0.16)
18:0 (stearic acid)	5.20 (2.46)	4.81 (2.20)	5.67 (2.67)
MUFAs			
16:1 (palmitoleic acid)	0.81 (0.39)	0.76 (0.36)	0.87 (0.42)
18:1 (oleic acid)	24.20 (9.82)	22.65 (8.95)	26.09 (10.49)
20:1 (eicosenoic acid)	0.50 (0.51)	0.48 (0.49)	0.52 (0.52)
22:1 (docosenoic acid)	0.21 (0.39)	0.21 (0.39)	0.21 (0.40)
PUFAs			
*n*-3 PUFAs			
18:3 (ALA)	1.58 (0.84)	1.54 (0.83)	1.64 (0.85)
18:4 (stearidonic acid)	0.04 (0.09)	0.04 (0.09)	0.04 (0.09)
20:5 (EPA)	0.03 (0.02)	0.02 (0.02)	0.03 (0.02)
22:5 (DPA)	0.06 (0.07)	0.06 (0.07)	0.06 (0.07)
22:6 (DHA)	0.18 (0.33)	0.19 (0.33)	0.18 (0.34)
*n*-6 PUFAs			
18:2 (LA)	9.67 (4.30)	9.08 (3.97)	10.40 (4.57)
20:4 (arachidonic acid)	0.11 (0.09)	0.10 (0.08)	0.12 (0.09)

ALA, alpha-linolenic acid; DHA, docosahexaenoic acid; DPA, docosapentaenoic acid; EPA, eicosapentaenoic acid; FA, fatty acid; g/d, grams per day; IQR, interquartile range; LA, linoleic acid; MCW, McCance and Widdowson; MUFA, monounsaturated fatty acid; PUFA, polyunsaturated fatty acid; SD, standard deviation; SFA, saturated fatty acid; USDA, US Department of Agriculture. Differences in mean FA intakes were compared by sex and education using the Kruskal–Wallis 1-factor ANOVA.

**Table 2 nutrients-14-03603-t002:** Comparison of total energy and major nutrient intakes between the main dataset (UKNDB) and the individual FA dataset (MCW + USDA) in 207,997 UK Biobank participants.

Nutrients	Main Dataset (UKNDB)				Individual FA Dataset (MCW + USDA)				Mean Difference ^1^	Percentage Difference ^2^
	Mean (SD)	Median (IQR)	5th Percentile	95th Percentile	Mean (SD)	Median (IQR)	5th Percentile	95th Percentile		
Total energy intake, kJ/d	8573 (2220)	8369 (2871)	5264	12587	8406 (2189)	8207 (2834)	5139	12371	−166.7	−1.94
Fat, g/d	72.4 (26.4)	69.5 (33.9)	34.4	120.1	70.6 (26.0)	67.8 (33.5)	33.2	117.7	−1.75	−2.42
SFAs, g/d	26.8 (11.3)	25.3 (14.4)	11.1	47.4	25.7 (10.9)	24.3 (14.0)	10.4	45.7	−1.09	−4.07
MUFAs, g/d	26.2 (10.2)	25.0 (12.9)	11.8	44.8	26.8 (10.6)	25.6 (13.5)	11.7	45.9	0.57	2.17
PUFAs ^3^, g/d	12.8 (5.5)	12.0 (6.7)	5.5	22.9	12.4 (5.1)	11.7 (6.3)	5.3	21.6	−0.43	−3.38
*n*-3 PUFAs ^4^, g/d	1.97 (0.97)	1.79 (1.12)	0.77	3.76	1.90 (1.01)	1.72 (1.24)	0.61	3.76	−0.08	−3.82
*n*-6 PUFAs ^5^, g/d	10.8 (4.9)	10.1 (5.8)	4.4	19.8	9.8 (4.3)	9.2 (5.3)	3.9	17.6	−1.06	−9.77
TFAs, g/d	1.18 (0.65)	1.08 (0.78)	0.34	2.37	1.04 (0.63)	0.93 (0.75)	0.26	2.20	−0.14	−11.96
Carbohydrates, g/d	252.7 (73.2)	246.8 (92.8)	143.7	382.4	247.9 (71.8)	242.3 (91.1)	140.6	374.3	−4.81	−1.90
Protein, g/d	80.3 (23.0)	78.5 (27.7)	46.2	120.2	79.6 (22.8)	77.8 (27.5)	45.8	119.2	−0.70	−0.88

kJ/d, kilojoules per day; TFA, trans fatty acid; UKNDB, UK nutrient databank. All mean differences were statistically significant from zero when using paired *t* tests or Wilcoxon’s rank sum test (*p* < 0.05). ^1^ Calculated as the difference of the mean (individual FA dataset—main dataset). ^2^ Calculated as the difference of the mean (individual FA dataset—main dataset) divided by main dataset and multiplied by 100. ^3^ For the main dataset this is the sum of *n*-3 and *n*-6 PUFAs. ^4^ For the individual FA dataset this is the sum of 18:3 (ALA), 18:4 (stearidonic acid), 20:5 (EPA), 22:5 (DPA) and 22:6 (DHA). ^5^ For the individual FA dataset this is the sum of 18:2 (LA) and 20:4 (arachidonic).

**Table 3 nutrients-14-03603-t003:** Intakes of individual FAs incorporated into the new individual FA dataset (MCW + USDA) in 207,997 UK Biobank participants.

Nutrients	Spearman’s *r*	Percentage in the Same Fifth	Percentage in the Same or Adjacent Fifth	Weighted *κ*
Total energy intake, kJ/d	0.996	91.2	100.0	0.945
Fat, g/d	0.987	85.3	99.9	0.907
SFAs, g/d	0.985	84.0	99.9	0.899
MUFAs, g/d	0.981	81.4	99.8	0.883
PUFAs ^1^, g/d	0.952	72.2	98.6	0.817
*n*-3 PUFAs ^2^, g/d	0.867	58.5	93.8	0.694
*n*-6 PUFAs ^3^, g/d	0.930	66.7	97.4	0.775
TFAs, g/d	0.892	62.6	94.9	0.729
Carbohydrates, g/d	0.993	88.5	100.0	0.928
Protein, g/d	0.984	83.3	99.9	0.895

^1^ For the main dataset this is the sum of *n*-3 and *n*-6 PUFAs. ^2^ For the individual FA dataset this is the sum of 18:3 (ALA), 18:4 (stearidonic acid), 20:5 (EPA), 22:5 (DPA) and 22:6 (DHA). ^3^ For the individual FA dataset this is the sum of 18:2 (LA) and 20:4 (arachidonic).

## Data Availability

Bona fide researchers can apply to use the UK Biobank dataset by registering and applying at http://ukbiobank.ac.uk/register-apply/ (accessed on 9 August 2022).

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
