# Peer review of "Evaluation of the New Individual Fatty Acid Dataset for UK Biobank: Analysis of Intakes and Sources in 207,997 Participants"

_nutrients, 2022, doi:10.3390/nu14173603_

Round 1

Reviewer 1 Report

The topic is important, and the development process is well written. This manuscript contains only the following minor errors.

1)     Line 29: SD should be spelt out.

2)     Line 107: There should be more explanation of how portion sizes are determined.

3)     Lines 123 and 164: The titles of these chapters are the same.

4)     Line 2935: These are the instruction for authors and should be deleted.

5)     Fig S1: Supplementary Fig 1 should be Fig S1

6)     Table S1: How were the fractions determined?

7)     Table S2Values obtained from USDA2’: 2 should be superscripted.

8)     Table S5: There is a typographical error in the subsidiary food group under ‘other bread’. ‘Fat speadsc’ may be ‘Fat spead3’. Please also check the footnote.

Reviewer 2 Report

The authors are commended for submitting a well-written manuscript reporting a newly created dataset focusing on individual fatty acid from UK Biobank. I find this study interesting and sheds light on a very important aspect of nutrition epidemiology – detailed individual FAs. The inter-rate agreement tests are generally valid, and all showed a substantial agreement suggest a great validity compared to original main data. Well written and insightful research with succinct discussion is strength of this paper.  

Author Response

We thank the reviewer for their positive comments. 

Reviewer 3 Report

This is an interesting and convenient work to update and improve the information for the intake of specific fatty acids (FA), among participants in the Biobank UK study and other prospective cohort studies. The manuscript is well written providing many details of the exhaustive work performed. The authors have completed the missing information on specific FA for the most relevant foods in the existing database of the Biobank UK, by using the MCW and USDA food composition tables, the two most exhaustive tables providing information for specific FAs. Thus, the manuscript is well justified and of scientific interest for the ongoing studies exploring the effects of main FA intake, and for comparative purposes among studies from other countries as well. I enclosed a few minor comments and suggestions. 

Specific comments

1. Lines 72-72: “Our secondary aim was to compare the estimation of major nutrient intakes between the new individual FA dataset and the main dataset”

This secondary objective should be better specified. Although latter were defined, major nutrients are undefined at this point of the manuscript. 

The same could be said for the “new individual FA dataset and the main dataset”. 

2. Lines  95-100: “The purpose of our analyses was to describe and compare nutrient intakes estimated in our sample as a whole (not individual participant intakes), and therefore we have included participants with only one 24-h dietary assessment”

It is unclear if this purpose is a new study aim or it is included in the first or secondary objective already. Please, clarify

3. Sections 2.4.1 and 2.4.2. were named the same as “Food code matching”. Is this correct?

4. As the authors recognized, the dietary assessment was poor for some particular food items (eg olive oil) which may limit future comparisons with other population studies carried out in other areas such as USA or Mediterranean countries.

On the other hand, a high proportion of participants did not provide information on diet or only provided one 24h recall. Using only one 24h recall may be also a limitation if the interest is classifying individuals to explore associations with the risk of disease because of the intraindividual variation for food and nutrient intakes. Although  this was not the purpose of this study, a more sensible  approach in epidemiological studies of big sample size like Biobank UK would be the use of validated Food Frequency Questionnaires as shown in other UK  prospective studies (eg. Norfolk EPIC cohort).

In this sense, it would be interesting to know if there any difference between mean nutrient daily intakes for individuals with only one 24-h recall and those with the average of 2-5 24-h?

At any rate, I suggest some comment in discussion or limitations on these issues.

5. Any explanation for the different main contributor to PUFAs in this study (LA 9.7 g/d) and those in the NHANES 2011-2012 study (17.4 g/d)?
